# RepDarkNet: A Multi-Branched Detector for Small-Target Detection in Remote Sensing Images

Liming Zhou [1,2] , Chang Zheng [1,2], Haoxin Yan [1,2], Xianyu Zuo [1,2,*], Yang Liu [1,2,3] , Baojun Qiao [1,2] and Yong Yang [4]

1 Henan Key Laboratory of Big Data Analysis and Processing, Henan University, Kaifeng 475000, China; lmzhou@henu.edu.cn (L.Z.); czheng@henu.edu.cn (C.Z.); 104754190933@henu.edu.cn (H.Y.); sea@vip.henu.edu.cn (Y.L.); qbj@henu.edu.cn (B.Q.)
2 School of Computer and Information Engineering, Henan University, Kaifeng 475000, China
3 Henan Province Engineering Research Center of Spatial Information Processing, Henan University, Kaifeng 475004, China
4 Institute of Plant Stress Biology, State Key Laboratory of Cotton Biology, Department of Biology, Henan University, Kaifeng 475000, China; 104752160097@vip.henu.edu.cn
* Correspondence: xianyu_zuo@henu.edu.cn

**Abstract:** Recent years have seen rapid progress in target-detection missions, whereas small targets, dense target distribution, and shadow occlusion continue to hinder progress in the detection of small targets, such as cars, in remote sensing images. To address this shortcoming, we propose herein a backbone feature-extraction network called "RepDarkNet" that adds several convolutional layers to CSPDarkNet53. RepDarkNet considerably improves the overall network accuracy with almost no increase in inference time. In addition, we propose a multi-scale cross-layer detector that significantly improves the capability of the network to detect small targets. Finally, a feature fusion network is proposed to further improve the performance of the algorithm in the $AP@0.75$ case. Experiments show that the proposed method dramatically improves detection accuracy, achieving $AP = 75.53\%$ for the Dior-vehicle dataset and $mAP = 84.3\%$ for the Dior dataset, both of which exceed the state-of-the-art level. Finally, we present a series of improvement strategies that justifies our improvement measures.

**Keywords:** deep learning; convolutional neural network; backbone network; target detection; remote sensing images

## 1. Introduction

A basic task in computer vision is target detection, which frames the region of interest in the input image in a bounding box. As space technology continues to develop, super-resolution remote sensing images are becoming increasingly important. However, target detection in remote sensing imaging remains a huge challenge and is thus receiving increasing research attention. According to different imaging bands, remote sensing can be divided into optical remote sensing, infrared remote sensing, SAR (Synthetic Aperture Radar), and other categories. Target detection in optical remote sensing images plays an important role in a wide range of applications such as environmental monitoring, geological hazard detection, land-use/land-cover (LULC) mapping, geographic information system (GIS) updating, precision agriculture, and urban planning [1].

In optical remote sensing imaging, targets usually have various orientations because of the differences in overhead views. Targets in optical remote sensing images thus consume a small number of target pixels distributed widely across different directions, making them easily affected by the surrounding environment. For example, vehicles in optical remote sensing images are often obscured by shadows, as shown in Figure 1. These factors make the difficult task of target detection in optical remote sensing images a research priority.

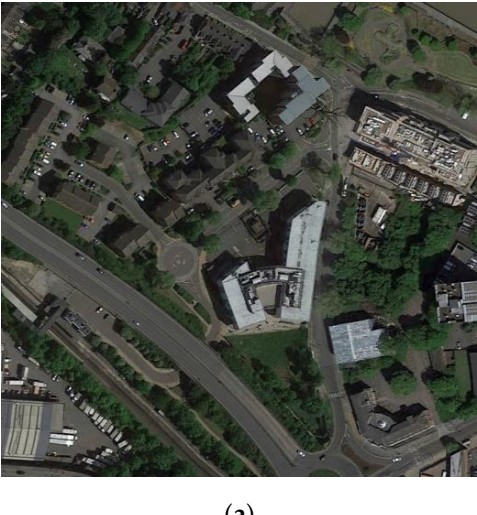
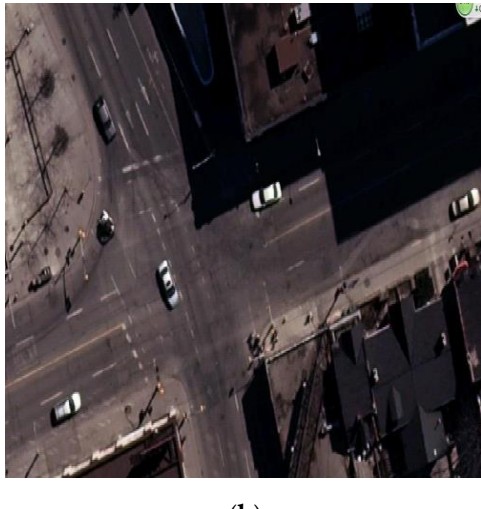

(**a**)        (**b**)

**Figure 1.** The vehicles in optical remote sensing imagery. Vehicles in (**a**) are generally below 10 pixels, and in (**b**) are obscured by shadows. Image in (**a**) is from the Dior dataset, and (**b**) from the HRRSD dataset.

In the last few years, the rise of deep learning has contributed greatly to the development of target detection. Good results have been achieved from a series of powerful target-detection algorithms, such as RCNN [2], Fast RCNN [3], Faster RCNN [4], SSD [5], YOLO [6,7], and ResNet [8]. However, the performance of these algorithms when applied to aerial-imaging tasks remains unsatisfactory.

In many convolutional network models, the backbone network is often used to extract target features, such as color, texture, and scale. The backbone network can provide several combinations of sensory field sizes and center steps to meet the target-detection requirements of different scales and classes [9]. Some commonly used backbone networks for various computer-vision tasks are VGG [10], ResNet [8], MobileNet [11–13], and DarkNet [6,7].

To detect small targets, the backbone network plays an important role. A good backbone network extracts more feature information, but feature information is more difficult to extract from small targets. However, in optical remote sensing imaging, car targets may consume fewer than 10 pixels because of the angle of the shot. To detect those targets in optical remote sensing images, we made the following propositions:

1. Influenced by RepVgg [14], we proposed a backbone network named "RepDarkNet" that combines training accuracy and detection speed. Experiments show that Rep-DarkNet performs better when applied to the Dior dataset than do YOLOv3 and YOLOv4 with DarkNet style as the backbone.
2. We proposed a cross-layer converged network for small targets in optical remote sensing images. The network contains multi-scale cross-layer detection and feature fusion networks.
3. Besides, larger input and GIoU [15] were used to improve very-small-target detection and tested separately by applying them to the Dior-vehicle dataset.

The remainder of this paper is organized as follows: Section 1 introduces related research on detection in optical remote sensing images. Section 2 describes the proposed approach to detect small targets. Section 3 describes the experimental design and experimental details, and Section 4 describes the experimental results and analysis. Finally, we conclude this study in Section 5.

## 2. Related Work

Usually, the horizontal bounding box (HBB) target-detection method is used for general scenes and for optical remote sensing target detection. Rabbi et al. [9] used GAN

(Generative Adversarial Networks) to convert images into super-resolution images and extract feature information from the images to improve small-target detection. Zhang et al. [16] proposed a hierarchical robust convolutional neural network (CNN) and built a large high-resolution target-detection dataset called HRSSD. Adam et al. [17] proposed YOLT, which evaluates satellite images of arbitrary size at a rate exceeding 0.5 km$^2$/s and locates small targets (5 pixels) at high resolution.

Optical remote sensing imagery contains mostly densely distributed targets of arbitrary orientations; therefore, HBB-based targets may overlap significantly. Oriented bracketing boxes are often used to detect rotating targets when characterizing targets in aerial images. Ding et al. [18] proposed a rotational region-of-interest learner to convert horizontal regions of interest to rotational regions of interest. Based on the rotation-sensitive regions, a rotation position-sensitive region alignment (RPS-RoI-Align) module was proposed to extract rotation-invariant features from rotation-sensitive regions and thereby facilitate subsequent classification and regression. Yang et al. [19] proposed a multi-class rotational detector SCRDet for small, randomly oriented, and intensively distributed targets in aerial imagery, which improves sensitivity to small targets through a sampling fusion network that fuses multilayer features with anchored sampling. Finally, Zhou et al. [20] proposed an anchorless polar-coordinate optical remote sensing target detector (P-RSDet) that provides competitive detection accuracy by using a simpler target representation model and fewer regression parameters.

Concerning vehicles in optical remote sensing imagery, numerous researchers have investigated the characteristics of car targets. Audebert et al. [21] trained a deep, fully convolutional network on ISPRS Potsdam and the NZAM/ONERA Christchurch datasets, and used the learned semantic graph to extract the exact vehicle segments, thereby detecting vehicles by simply extracting connected parts. Zhang et al. [22] proposed a YOLOv3-based deeply separable attention-guided network for the real-time detection of small vehicle targets in optical remote sensing imagery. Shi et al. [23] proposed a single-stage anchorless detection method to detect vehicles in arbitrary directions. This method transforms vehicle detection into a multitask learning problem that requires the use of a full convolutional network to directly predict high-level vehicle features.

## 3. Materials and Methods

This section introduces the backbone network RepDarkNet, which is a multi-scale cross-layer detector and feature fusion network. In addition, we tested some common methods to improve the detection rate of small targets.

### 3.1. Reasons for Choosing DarkNet Style

We chose the DarkNet structure because it combines accuracy and speed. Currently, backbone networks are carefully designed to include a multi-branch structure, such as MobileNet [11–13], Inception [24], or DenseNet [25]. They have common characteristics; for instance, DenseNet makes the topology more complex by connecting lower layers to many higher layers, and while it offers a high-performance conversion network, it does so at the cost of a mass of GPUs.

Darknet53 is a simple two-branch network that adds branches in a top-down convolution process. It is mainly composed of a series of fully designed residual blocks. As can be seen in Figure 2, the residual block loads the output of the two convolutions plus the output of a skip connection together as the input information for the next layer of convolution. The elaborate network layer of DarkNet produces surprisingly fast detection.

In a multi-branch network structure, the paths between branches are not strongly interdependent, so the structure can be seen as a collection of many paths [26]. On a small scale, the DarkNet-style network is composed of multiple residual block results, whereas on a larger scale, it can be seen as five layers, as in Figure 2, where we divide the backbone network into a five-block structure (*R*1–*R*5). In the field of target detection, deeper network structures tend to be stronger than shallower structures, although building

multiple branching structures in each residual layer increases training time and detection time and wastes significant computational resources in return for poor accuracy. Therefore, implementing such network structures in one big module is a good way to improve them. YOLOv4 thus makes a breakthrough in accuracy, so we improve on the backbone network CSPDarkNet.

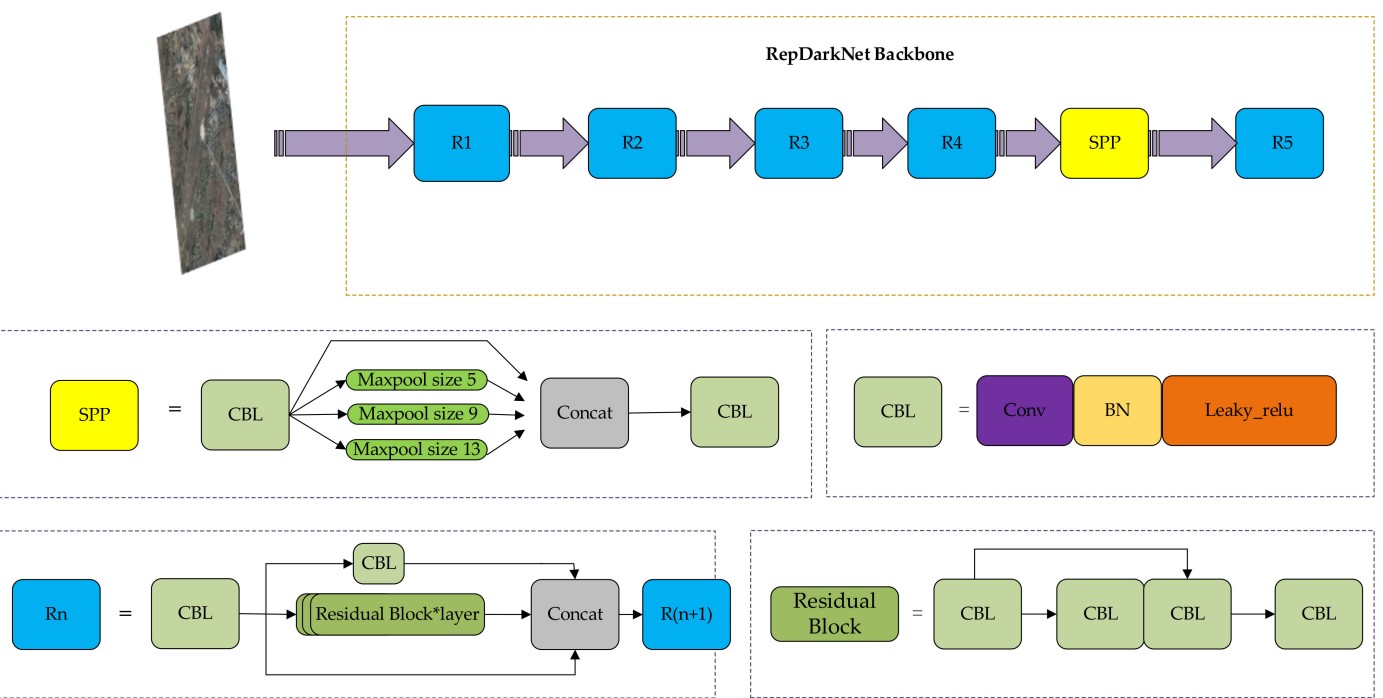

**Figure 2.** Details of backbone network. In Rn, when $n = 1$, layer = 1; when $n = 2$, layer = 2; when $n = 3$ or 4, layer = 8; and when $n = 5$, layer = 4. The purple arrow represents the connection between $R_n$ and $R_{n+1}$ blocks, the output information of $R_n$ is the input information of $R_{n+1}$.

### 3.2. Overview of DarkNet-Style Backbone Network

The proposed RepVGG (re-parameterization VGG) [14] makes VGG shine again in classification tasks. It is a simple, yet powerful, neural network structure. For target detection, DarkNet is an excellent and fast framework, but accuracy is what many researchers now focus on. In this work, we continue the simple structure of DarkNet and design a muti-branch backbone network RepDarkNet (re-plan the branching structure of DarkNet), the details of which are given in Figure 2, where $R_n$ represents a four-branch module in RepDarkNet. In the SPP module, the three maxpool layers are of sizes 5, 9, and 13, respectively.

CSPDarkNet is one of the backbone networks of the CSP(Cross-Stage-Partial-connections) module [7]. If CSPDarkNet is a two-branch structure, RepDarkNet is a simple, but powerful, three-branch structure. Compared with CSPDarkNet, it adds only one layer of branches per layer and constitutes a module with the information $y = f(x) + g(x) + x$, where $f(x)$ can be expressed as:

$$f_i(x) = W_i \cdot \delta(B(x)), \tag{1}$$

where $x$ is the output of the previous layer, $W$ is the weight information, $\cdot$ indicates convolution, B is the batch normalization operation, and $\delta(x) = \max(x, 0)$ is the Relu function. To explain $g(x)$, we take as an example R2, which contains two residual modules, each of which can be expressed as:

$$y_i = f_i(y_{i-1}) + y_{i-1}. \tag{2}$$

Given an input value $y_0$, then the first residual output is:

$$y_3 = y_2 + f_3(y_2) \tag{3}$$

$$= [y_1 + f_2(y_1)] + f_3(y_1 + f_2(y_1)) \tag{4}$$

$$= [y_0 + f_1(y_0) + f_2(y_0 + f_1(y_0))] + f_3(y_0 + f_1(y_0) + f_2(y_0 + f_1(y_0))). \tag{5}$$

Similarly, the second convolution result is $g(x) = y_6$, so:

$$y_6 = y_5 + f_6(y_5) \tag{6}$$

$$= [y_4 + f_5(y_4)] + f_6(y_4 + f_5(y_4)) \tag{7}$$

$$= [y_3 + f_4(y_3) + f_5(y_3 + f_4(y_3))] + f_6(y_3 + f_4(y_3) + f_5(y_3 + f_4(y_3))). \tag{8}$$

In the process of convolution, CNNs gradually lose the feature information of small targets, whereas the multi-branch module is built to send the upper-layer information directly to the deep layer, enriching the feature information of the deep layer network, especially for small targets. The experimental results also show that RepDarkNet detects small targets better than other algorithms do.

### 3.3. Cross-Layer Fusion Network

The cross-layer fusion network consists of two parts: a multi-scale cross-layer detector and a neck feature fusion network.

In a YOLOv4 network, the YOLO head usually splits the image into $19 \times 19$, $38 \times 38$, and $76 \times 76$ grids. However, in optical remote sensing images, small targets are usually smaller than 30 or even 20 pixels; therefore, adding a detection head at a shallow level is a good way to improve the detection accuracy of small targets, and studies show that this is feasible [27]. To detect small targets, as shown in Figure 3, N3 does not work well. Segmenting a $1080 \times 1080$ pixel image into a $76 \times 76$ grid makes for difficult detection based on the existing feature information. Therefore, a multi-scale cross-layer detector was designed that takes into account the size of the network and the inference time. The detector uses only layers N1, N2, and N4 and divides the image into three scales of $19 \times 19$, $38 \times 38$, and $152 \times 152$ to detect large and small targets in the image during inference. Experiments show that the multi-scale cross-layer detector is much more effective, especially in detecting small targets. Figure 4 shows its network structure.

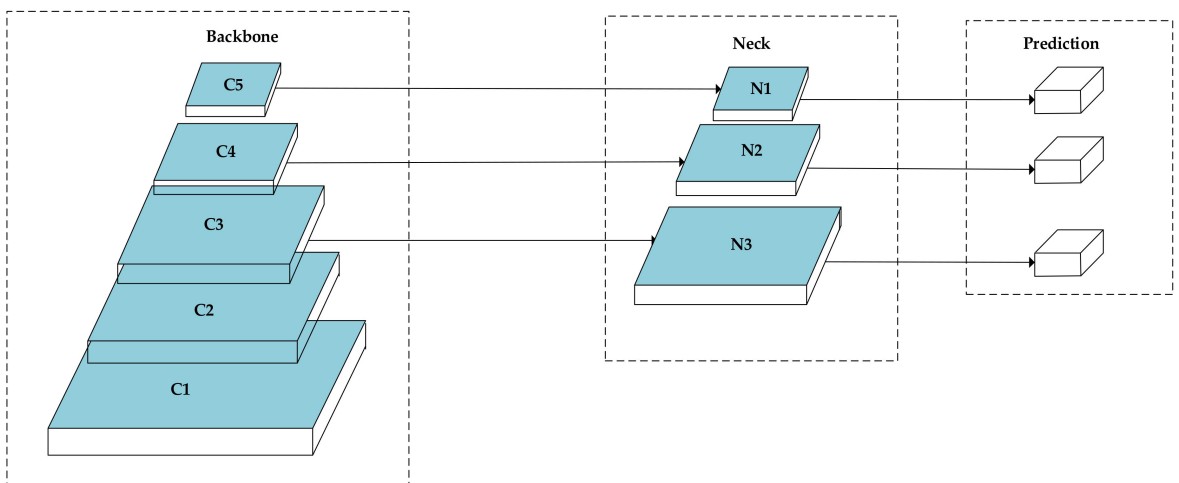

**Figure 3.** Overviews of multi-scale detection in YOLOv4.

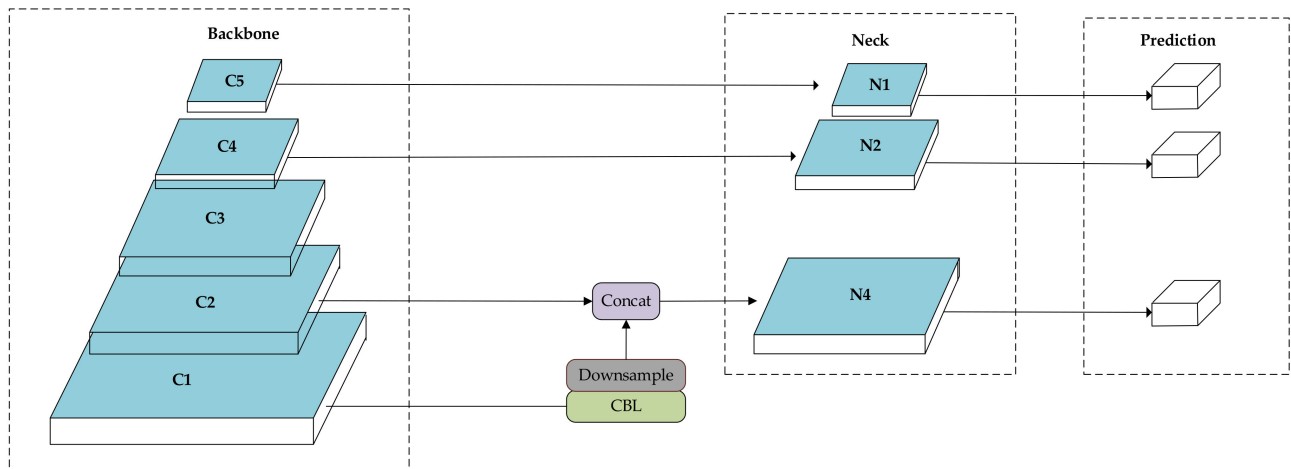

**Figure 4.** Overview of multi-scale cross-layer detectors and feature fusion networks.

Inspired by Lim et al. [28], we fused higher-level feature maps from the target feature layer to provide context for a given feature map of the target to be detected. For example, in RepDarkNet, given the target features from *R*2, the contextual features come from the *R*1 layer, as shown in Figure 4. However, *R*1 and *R*2 have different dimensions, so the network channels are downscaled by downsample in order to ensure that the size of the feature information is the same on both sides before fusing the features, and the fused feature information is convolved for detection by a multi-scale cross-layer detector.

*3.4. Options for Improving Accuracy of Small-Target Detection*
3.4.1. GIoU

As a loss of the Bbox regression, GIoU [15] (Generalized Intersection over Union) is always less than or equal to IoU [29] (Intersection over Union), where $0 \leq \text{IoU} \leq 1$ and $-1 \leq \text{GIoU} \leq 1$. When the two shapes coincide exactly, GIoU = IoU = 1. GIoU and IoU can be expressed as:

$$\text{GIoU} = \text{IoU} - \frac{|C/(A \cup B)|}{|C|}, \tag{9}$$

$$\text{IoU} = \frac{|A \cap B|}{|A \cup B|}, \tag{10}$$

where *A* and *B* are arbitrary shapes. In this algorithm, *A* and *B* identify the precision $= \frac{TP}{TP+FP}$ marker box in the image, and the algorithm predicts the anchor selection box. *C* is the minimum closed shape.

Both IoU and GIoU serve as distances between *A* and *B*; if $|A \cap B| = 0$, then IoU(*A*, *B*) = 0, so IoU does not indicate whether the two shapes are nearby or far-away.

3.4.2. Larger Input Size

Larger input size increases the size of the target area and makes it easier to preserve the features of smaller targets. However, increasing the input size takes up more memory during training. In our experiments, we increased the maximum input size only from $608 \times 608$ to $640 \times 640$ because of experimental constraints. We could also increase the input size while decreasing the batch value, although this would significantly increase the training time.

## 4. Experiments, Results, and Discussion

This section introduces the dataset that we used, the evaluation criteria, the details of the experiments, and the results. The experimental environment for this work is based on the Ubuntu 16.04 operating system. The hardware platform was an Intel(R) Xeon(R) Silver 4114 CPU @ 2.2 GHz and a Quadro P4000 8 GB * 2 GPU.

In the process of training, the momentum is set to 0.949, initial learning rate is set to 0.0013, and batch is set to 64. There are different numbers of iterations for different datasets for training. In the Dior-vehicle dataset, when the number for training iterations is 3200 and 3600, the learning rate is adjusted to 0.00013 and 0.000013, respectively. We have iterated 4000 times of this algorithm. Finally, in this hardware environment, the training time is about 12 h.

### 4.1. Dataset

Dior [30] and NWPU VHR-10 [30] were the datasets used to test our method. To show more intuitively the results of the algorithm to detect small targets, we tested the Dior-vehicle [31] dataset using the ablation experiment method.

### 4.1.1. Dior-Vehicle Dataset

The Dior-vehicle is a separate class of vehicle in Dior; the dataset has mostly small targets, and car targets consuming fewer than 10 pixels are flagged. Here, we follow COCO [32] for object-size classification, which small objects area is less than $32 \times 32$ pixels. The Dior-vehicle dataset has 6421 images and over 32,000 targets. In our experiments, we divide the training and test sets in the ratio 6:4. We then test the feasibility of the proposed method via an ablation experiment [4].

### 4.1.2. Dior Dataset

Dior is a large-scale benchmark dataset for target detection in optical remote sensing images consists of 23,463 images and 190,288 instances. It contains 20 categories. In the experiments reported herein, we use images from "trainval" for training and images from test for testing. The "trainval" is a TXT document. The author of the Dior dataset wrote the name of the image which is used for training into this document.

Besides, the size of objects varies over a wide range, not only in terms of spatial resolution, but also in terms of inter- and intra-class size variation between objects. Next, the images were obtained under different imaging conditions, weather, seasons, and image quality, so there are large differences in the targets therein. Some images may have noise and some targets in motion would be blurred. Finally, it has a high degree of inter-class similarity and intra-class diversity. These characteristics greatly increase the difficulty of detection.

If you want to find out more information about the dataset, please visit http://www.escience.cn/people/gongcheng/DIOR.html, accessed on 7 October 2020.

### 4.1.3. NWPU VHR-10 Dataset

The NWPU VHR-10 dataset is a research-only public dataset containing 10 classes. NWPU VHR-10 has 650 positive optical images and 150 negative optical images. In our experiments, we divided the 650 positive optical images with annotations at a 5:5 rate to produce a training set and a test set. If you want to find out more information about the dataset, please http://www.escience.cn/people/gongcheng/NWPU-VHR-10.html (accessed on 10 May 2021).

### 4.2. Evaluation Standards

The F1 score, precision, recall, and IoU are also considered in our measurement in the ablation experiment to test the Dior-vehicle dataset. The formula for IoU is given in Equation (1), and the F1 score, precision, and recall can be expressed by:

$$\text{precision} = \frac{TP}{TP + FP}, \tag{11}$$

$$\text{recall} = \frac{TP}{TP + FN}, \tag{12}$$

$$\text{F1 score} = \frac{2(\text{precision} \times \text{recall})}{\text{precision} + \text{recall}}. \tag{13}$$

$TP$, $FP$, and $FN$ are the number of true positives, false positives, and false negatives, respectively. In this paper, we use $AP$ (average precision) and $mAP$ (mean average precision) to evaluate all methods. These can be expressed as follows:

$$AP_i = \int_0^1 \text{precision}_i(r_i)d(r_i), \tag{14}$$

$$mAP = \frac{1}{n}\sum_{i=1}^{n} AP_i. \tag{15}$$

When testing the results, a threshold value for IoU is set, such as 0.5 or 0.75. When the detected box and the actual annotated box exceed this threshold, they are considered true positives; otherwise, they are false positives. When the detected box is not present in the actual annotation, they are false negatives.

### 4.3. Experimental Results and Analyses

### 4.3.1. Result of Dior-Vehicle and Analyses

To investigate the impression of the method we used on the detection results, we used an ablation experiment to test the dataset. As shown in Table 1, "a" represents the Rep backbone, "b" the larger input, "c" the GIoU, "d" the multi-scale cross-layer detector, and "e" the feature fusion network. For example, RepDarkNet-A2 uses both "a" and "c". RepDarknet-B3 uses all the methods and is our final algorithm.

**Table 1.** Improvement options and comparison of results.

|  | a | b | c | d | e | AP@0.5(%) | AP@0.75(%) |
|---|---|---|---|---|---|---|---|
| YOLOv4 | - | - | - | - | - | 60.33 | 26.35 |
| RepDarkNet-A0 | √ | - | - | - | - | 71.58 | 34.16 |
| RepDarkNet-A1 | √ | √ | - | - | - | 74.51 | 38.51 |
| RepDarkNet-A2 | √ | - | √ | - | - | 74.16 | 37.10 |
| RepDarkNet-B1 | √ | - | - | √ | - | 75.82 | 33.32 |
| RepDarkNet-B2 | √ | - | - | √ | √ | 74.87 | 35.37 |
| RepDarkNet-B3 | √ | √ | √ | √ | √ | 75.52 | 38.40 |

In the first step, the RepDarkNet-A0, we add only the RepDarkNet backbone on top of the YOLOv4 algorithm. As can be seen in Table 1, the $AP$ is greatly improved (60.33–71.58%). Next, we use larger input and RepDarkNet backbone, and the $AP$ is increased by 2.93%. Thirdly, we use the GIoU and RepDarkNet backbone, and compared to the RepDarkNet-A0, the $AP$ is increased by 2.58%. Then, we use RepDarkNet backbone and the multi-scale cross-layer detector, and compared to the RepDarkNet-A0, the $AP$ is increased by 4.24%. This is the highest promotion method other than RepDarkNet backbone. In step five, we use the RepDarkNet backbone, the multi-scale cross-layer detector and the feature fusion network. Compared to RepDarkNet-B1, $AP@50$ is 0.95% less, but $AP@75$ is 2.05% more. Finally, we added all the methods. The proposed method approaches the best performance for the Dior-vehicle dataset in both the $AP@50$ and $AP@75$ cases. Figure 5 shows the comparison of loss and $mAP$ between the YOLOv4 and RepDarkNet-B3. It can be seen that RepDarkNet-B3's loss value is lower than YOLOv4 after 1200 iterator times. At the same time, the $mAP$ value is higher than YOLOv4. Besides, Figure 6 shows the variation in $AP@50$ and $AP@75$ for each method.

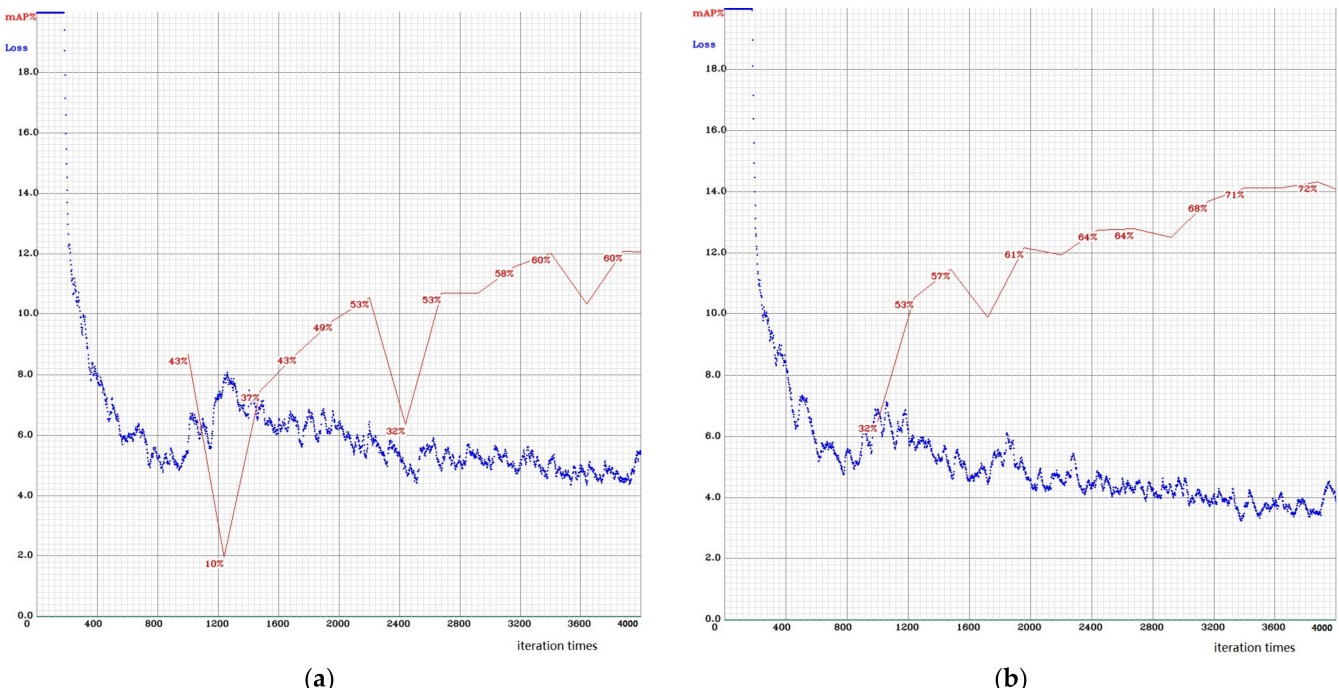

**Figure 5.** YOLOv4 (**a**) and RepDarkNet-B3 (**b**) loss comparison.

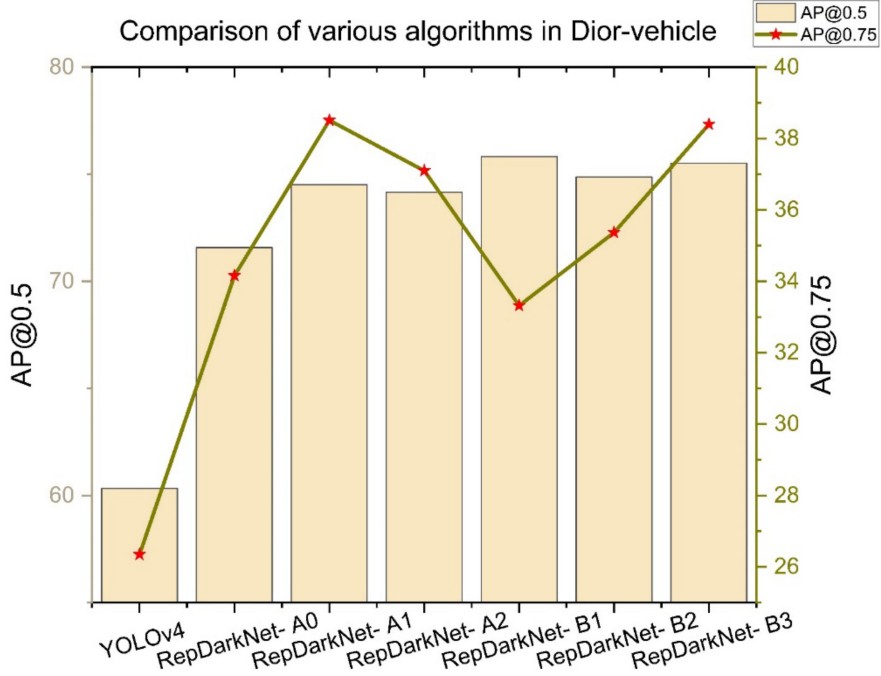

**Figure 6.** Comparison of various algorithms in Dior-vehicle.

As can be seen in Figure 7(a1,a2,b1,b2), the YOLOv4 algorithm incorrectly detects the ship as a car, whereas RepDarkNet accurately frames the car targets of the images. It is clear from Figure 7(a3,a4,b3,b4) that RepDarkNet detects small targets much better than YOLOv4 does. The results in Tables 1 and 2 demonstrate the clear improvement offered by the proposed method for small targets. Figure 7(a5,a6) show that the YOLOv4 algorithm incorrectly detects road signs as cars, whereas RepDarkNet correctly ignores the road signs. Considering the inconspicuous markings of the misdetection in Figure 7(a5,a6), we have marked it with a yellow circle in order for the reader to notice this information more quickly. This demonstrates that RepDarkNet is more robust than YOLOv4.

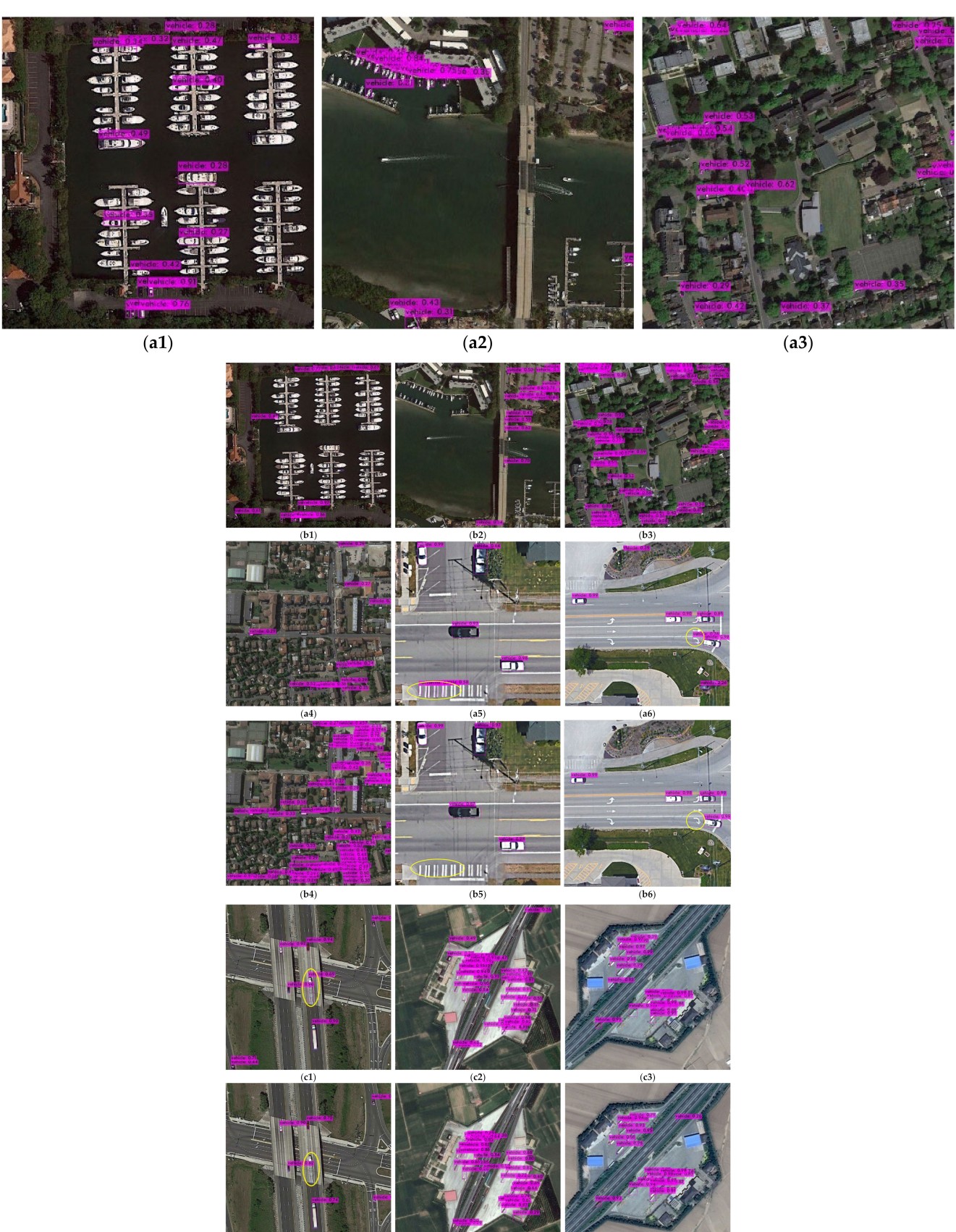

**Figure 7.** Comparison of detection results between YOLOv4 (**a1–a6,c1–c3**) and RepDarkNet (**b1–b6,d1–d3**).

**Table 2.** Comparison of recall, precision, F1-score, and IoU by methods.

| Method | F1-Score | Recall | Precision | IoU |
|:---:|:---:|:---:|:---:|:---:|
| YOLOv4 | 58% | 59% | 57% | 42.90% |
| RepDarkNet-A0 | 69% | 68% | 71% | 53.89% |
| RepDarkNet-A1 | 71% | 72% | 70% | 53.84% |
| RepDarkNet-A2 | 72% | 72% | 71% | 54.76% |
| RepDarkNet-B1 | 70% | 75% | 66% | 49.38% |
| RepDarkNet-B2 | 72% | 72% | 72% | 54.75% |
| RepDarkNet-B3 | 72% | 70% | 77% | 59.46% |

Furthermore, there are more special categories of vehicles, such as lorries, characteristics of which are different from those of ordinary cars. In the dataset, although lorries are also labeled, the number of lorries in the data is relatively much smaller. As shown in Figure 7(c1,d1), the detection accuracy of our proposed algorithm is higher than that of YOLOv4 for lorries.

Of course, our algorithm also has some shortcomings. By looking at Figure 7(c2,c3,d2,d3), we can see that it is easy to mistake containers as cars, regardless of RepDarkNet or YOLOv4. This is a problem we want to solve in our future research.

Finally, to test the generality of the proposed detector, we directly detect the vehicle class in the HRRSD dataset [16] without any training using Dior's weights. The final detection result of $AP = 50.73\%$ was obtained. Although far from expected, this far exceeds the result of $AP = 23.78\%$ obtained by YOLOv4. Of course, we also hope that in future research, there is a greater breakthrough in the generality performance of the detector.

4.3.2. Results of the Dior Dataset and Analyses

As shown in Table 3, to better display the results, we number each category in the dataset.

**Table 3.** Categories in the DIOR dataset and their corresponding numbers.

| c1 | Airplane | c6 | Chimney | c11 | Ground track field | c16 | Storage tank |
|:---:|:---:|:---:|:---:|:---:|:---:|:---:|:---:|
| c2 | Airport | c7 | Dam | c12 | Harbor | c17 | Tennis court |
| c3 | Baseball field | c8 | Expressway service area | c13 | Overpass | c18 | Train station |
| c4 | Basketball court | c9 | Expressway toll station | c14 | Ship | c19 | Vehicle |
| c5 | Bridge | c10 | Golf court | c15 | Stadium | c20 | Windmill |

Table 4 shows clearly that the proposed algorithm performs the best for the Dior dataset, far outperforming other algorithms for the detection of small targets, such as cars or ships. For relatively large targets, such as bridges or basketball courts, the detection accuracy is also improved slightly relative to other algorithms, which demonstrates the strong applicability of the proposed algorithm. Compared with the YOLOv4 algorithm, most classes in our test are more accurate, which demonstrates convincingly that RepDarkNet is an excellent target-detection algorithm.

Figure 8a–i shows visualization results of the proposed method applied to the Dior dataset. RepDarkNet performs well not only for small targets such as ships, planes, and storage tanks in the dataset but also for other classes such as tennis and basketball courts. The results in Table 4 also show that in addition to dam, expressway toll station, tennis court and train station, the proposed approach produces the best results compared with other algorithms. Only in dam, expressway toll station and tennis court, YOLOv4 is better than RepDarkNet.

**Table 4.** Comparison of the results of the algorithms on the Dior dataset.

| Method | Backbone | c1 | c2 | c3 | c4 | c5 | c6 | c7 | c8 | c9 | c10 | c11 | c12 | c13 | c14 | c15 | c16 | c17 | c18 | c19 | c20 | mAP |
|---|---|---|---|---|---|---|---|---|---|---|---|---|---|---|---|---|---|---|---|---|---|---|
| SSD [30] | VGG16 | 59.5 | 72.7 | 72.4 | 75.7 | 29.7 | 65.8 | 56.6 | 63.5 | 53.1 | 65.3 | 68.6 | 49.4 | 48.1 | 59.2 | 61.0 | 46.6 | 76.3 | 55.1 | 27.4 | 65.7 | 58.6 |
| YOLOv3 [30] | Darknet-53 | 72.2 | 29.2 | 74.0 | 78.6 | 31.2 | 69.7 | 26.9 | 48.6 | 54.4 | 31.1 | 61.1 | 44.9 | 49.7 | 87.4 | 70.6 | 68.7 | 87.3 | 29.4 | 48.3 | 78.7 | 57.1 |
| Faster RCNNwith | ResNet-50 | 54.1 | 71.4 | 63.3 | 81.0 | 42.6 | 72.5 | 57.5 | 68.7 | 62.1 | 73.1 | 76.5 | 42.8 | 56.0 | 71.8 | 57.0 | 53.5 | 81.2 | 53.0 | 43.1 | 80.9 | 63.1 |
| FPN [30] | ResNet-101 | 54.0 | 74.5 | 63.3 | 80.7 | 44.8 | 72.5 | 60.0 | 75.6 | 62.3 | 76.0 | 76.8 | 46.4 | 57.2 | 71.8 | 68.3 | 53.8 | 81.1 | 59.5 | 43.1 | 81.2 | 65.1 |
| Mask-RCNNwith | ResNet-50 | 53.8 | 72.3 | 63.2 | 81.0 | 38.7 | 72.6 | 55.9 | 71.6 | 67.0 | 73.0 | 75.8 | 44.2 | 56.5 | 71.9 | 58.6 | 53.6 | 81.1 | 54.0 | 43.1 | 81.1 | 63.5 |
| FPN [30] | ResNet-101 | 53.9 | 76.6 | 63.2 | 80.9 | 40.2 | 72.5 | 60.4 | 76.3 | 62.5 | 76.0 | 75.9 | 46.5 | 57.4 | 71.8 | 68.3 | 53.7 | 81.0 | 62.3 | 43.0 | 81.0 | 65.2 |
| RetinaNet [30] | ResNet-50 | 53.7 | 77.3 | 69.0 | 81.3 | 44.1 | 72.3 | 62.5 | 76.2 | 66.0 | 77.7 | 74.2 | 50.7 | 59.6 | 71.2 | 69.3 | 44.8 | 81.3 | 54.2 | 45.1 | 83.4 | 65.7 |
|  | ResNet-101 | 53.3 | 77.0 | 69.3 | 85.0 | 44.1 | 73.2 | 62.4 | 78.6 | 62.8 | 78.6 | 76.6 | 49.9 | 59.6 | 71.1 | 68.4 | 45.8 | 81.3 | 55.2 | 44.4 | 85.5 | 66.1 |
| PANet [30] | ResNet-50 | 61.9 | 70.4 | 71.0 | 80.4 | 38.9 | 72.5 | 56.6 | 68.4 | 60.0 | 69.0 | 74.6 | 41.6 | 55.8 | 71.7 | 72.9 | 62.3 | 81.2 | 54.6 | 48.2 | 86.7 | 63.8 |
|  | ResNet-101 | 60.2 | 72.0 | 70.6 | 80.5 | 43.6 | 72.3 | 61.4 | 72.1 | 66.7 | 72.0 | 73.4 | 45.3 | 56.9 | 71.7 | 70.4 | 62.0 | 80.9 | 57.0 | 47.2 | 84.5 | 66.1 |
| CF2PN [33] | VGG16 | 78.3 | 78.3 | 76.5 | 88.4 | 37 | 71 | 59.9 | 71.2 | 51.2 | 75.6 | 77.1 | 56.8 | 58.7 | 76.1 | 70.6 | 55.5 | 88.8 | 50.8 | 36.9 | 80.4 | 67.3 |
| MFPnet [34] | VGG16 | 76.6 | 83.4 | 80.6 | 82.1 | 44.3 | 75.6 | 68.5 | 85.9 | 63.9 | 77.3 | 77.2 | 62.1 | 58.8 | 77.2 | 76.8 | 60.3 | 86.4 | 64.5 | 41.5 | 80.2 | 71.2 |
| CANet [35] | ResNet-101 | 70.3 | 82.4 | 72 | 87.8 | 55.7 | 79.9 | 67.7 | 83.5 | 77.2 | 77.3 | 83.6 | 56.0 | 63.6 | 81.0 | 79.8 | 70.8 | 88.2 | 67.6 | 51.2 | 89.6 | 74.3 |
| YOLOv4 | CSPDarkNet | 96.0 | 87.9 | 94.7 | 91.9 | 60.0 | 90.8 | **69.9** | 92.1 | **87.5** | 87.6 | 83.7 | 55.6 | 68.7 | 94.6 | 83.8 | 88.4 | **95.7** | 44.4 | 62.1 | 90.4 | 81.3 |
| ours | RepDarkNet | **97.7** | **89.5** | **94.9** | **92.5** | **62.2** | **91.4** | 68.2 | **94.5** | 85.8 | **87.1** | **91.4** | **62.9** | **72.9** | **95.7** | **92.3** | **89.5** | 95.5 | 56.7 | **71.4** | **93.6** | **84.3** |

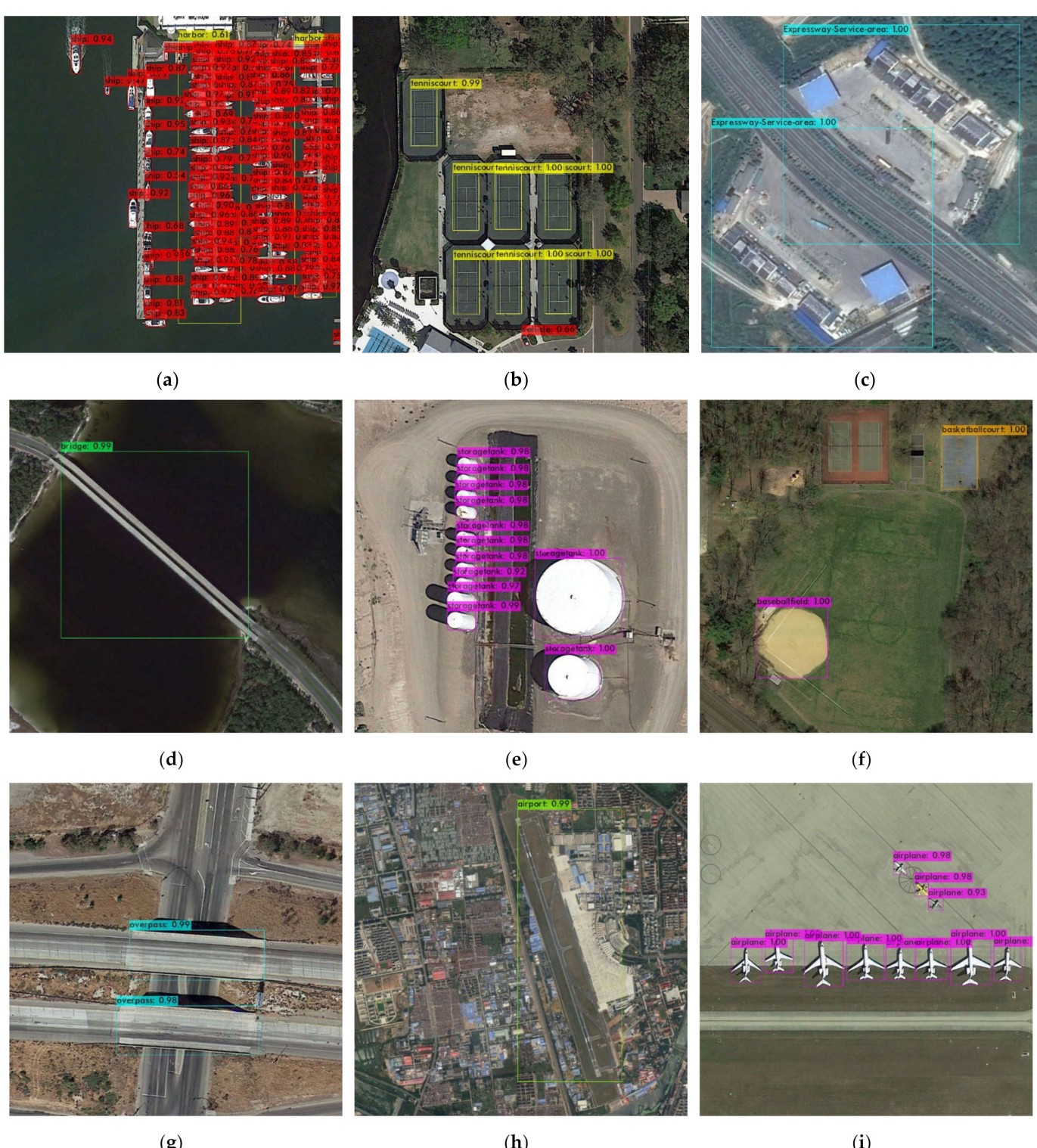

**Figure 8.** Detection results of RepDarkNet in the Dior dataset (**a–i**).

### 4.3.3. Results of NWPU VHR-10 and Analyses

Table 5 and Figure 9a–i illustrate the performance of RepDarkNet compared with those of other published methods when applied to the NWPU VHR-10 dataset. We first introduce the abbreviations of classes in the Table 5. The categories are Airplane (PL), Ship (SP), Storage tank (ST), Baseball diamond (BD), Tennis court (TC), Basketball court (BC), Ground track field (GT), Harbor (HB), Bridge (BR), and Vehicle (VH).

**Table 5.** Comparison of results from the NWPU VHR-10 dataset.

| Method | AP (%) for Each Target Category | | | | | | | | | | mAP (%) |
| | PL | SH | ST | BD | TC | BC | GT | HA | BR | VE | |
|---|---|---|---|---|---|---|---|---|---|---|---|
| SSD512 [36] | 90.40 | 60.90 | 79.80 | 89.90 | 82.60 | 80.60 | 98.30 | 73.40 | 76.70 | 52.10 | 78.40 |
| SAPNet [36] | 97.80 | 87.60 | 67.20 | 94.80 | 99.50 | 99.50 | 95.90 | 96.80 | 68.00 | 85.10 | 89.20 |
| StAN-Enh [36] | 94.80 | 79.10 | 98.20 | 96.70 | 89.10 | 89.60 | 93.50 | 91.00 | 62.70 | 93.80 | 88.9 |
| CANet [35] | 100.0 | 81.9 | 94.6 | 90.3 | 90.7 | 90.6 | 99.8 | 89.8 | 93.9 | 89.9 | 92.2 |
| YOLOv4 | 99.99 | 81.33 | 98.80 | 97.41 | 97.40 | 95.52 | 99.37 | 82.27 | 77.30 | 93.70 | 92.31 |
| ours | 99.96 | **93.71** | 98.07 | **97.47** | **99.56** | 99.17 | 99.59 | 84.94 | 74.24 | **94.29** | **94.1** |

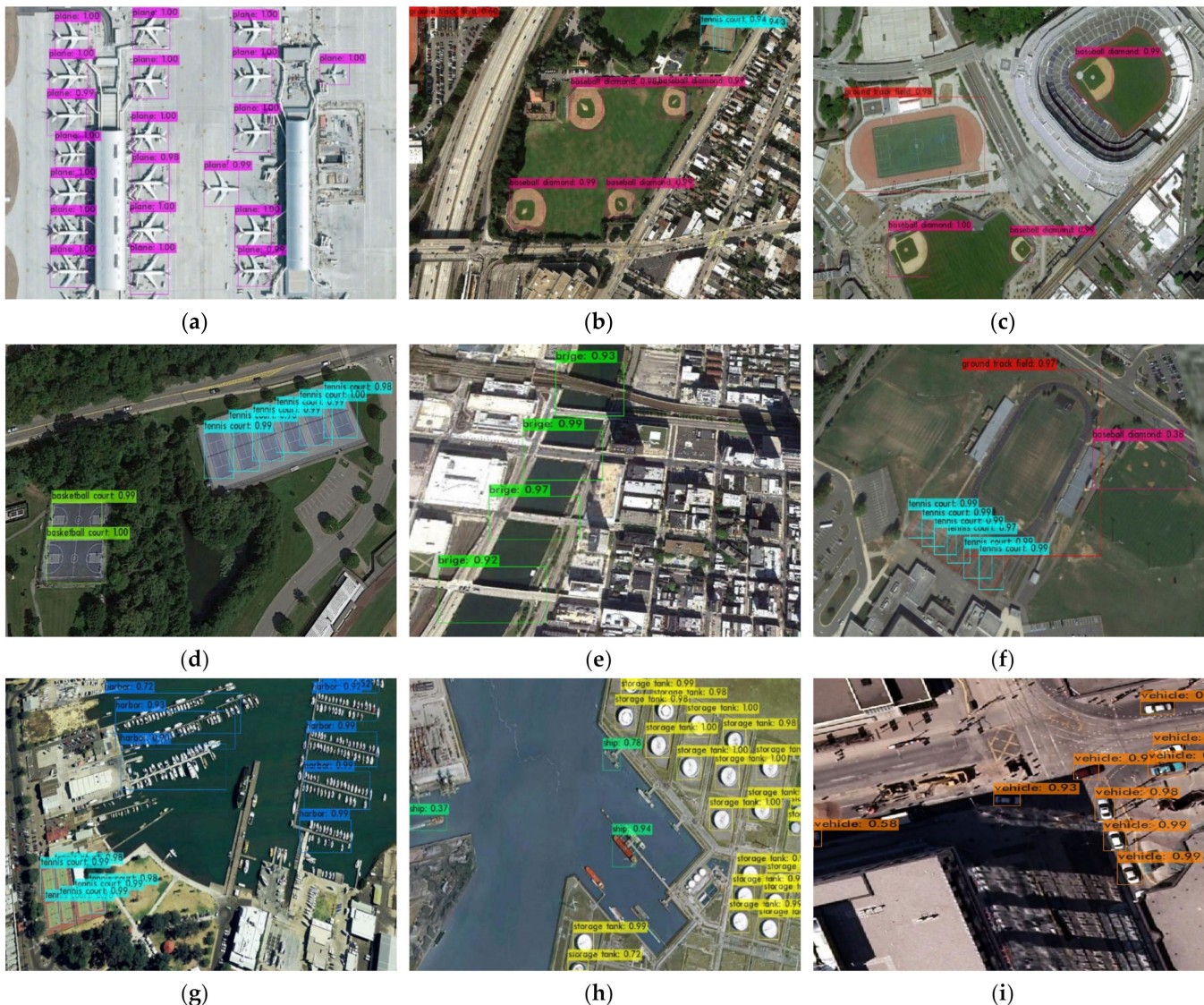

**Figure 9.** RepDarkNet detection results in the NWPU VHR-10 dataset (**a**–**i**).

The proposed method produces $mAP = 94.1\%$, which is the best performance of all the methods. Note that no small targets appear in the dataset and that the classes SH, BD, TC, and VE which are bolded in Table 5 have the highest accuracy in the experimental results. Overall, $mAP$ is the highest too, which is good evidence of the strong applicability of our method.

## 5. Conclusions

We propose a new backbone feature-extraction network called "RepDarkNet" that provides improved target detection by considering network throughput and computation time. In addition, for small targets in the vehicle class of optical remote sensing images, we propose a multi-scale cross-layer detector and feature fusion network. Finally, in experiments, RepDarkNet achieves $AP@0.5 = 75.52\%$ and $AP@0.75 = 38.4\%$, both of which are near-optimal, and a series of ablation experiments justify our approach. In extended experiments, RepDarkNet running under Quadro P4000 obtains $mAP = 84.3\%$ when applied to the Dior dataset and $mAP = 94.1\%$ when applied to the NWPU VHR-10 dataset, outperforming other algorithms and demonstrating the broad applicability of the proposed algorithm. To test the algorithm for generalized rows, we obtained $AP = 50.73\%$ performance in HRRSD dataset using the training weights of the Dior-vehicle dataset with no other additional training. Under the same conditions, YOLOv4 obtains $AP = 23.78\%$ performance. Thus, in terms of generality, RepDarkNet is much better than YOLOv4. By optimizing and evaluating the proposed small-target network models under three datasets (Dior-vehicle, Dior, and NWPU VHR-10), these results have practical implications for improving detection techniques based on optical remote sensing images.

**Author Contributions:** Conceptualization, Liming Zhou and Chang Zheng; Methodology, Liming Zhou and Chang Zheng; Software, Haoxin Yan; Validation, Liming Zhou, Chang Zheng and Xianyu Zuo; Writing—original draft, Liming Zhou and Chang Zheng; Writing—review & editing, Chang Zheng, Yang Liu, Baojun Qiao and Yong Yang. All authors have read and agreed to the published version of the manuscript.

**Funding:** This work is supported by grants from National Basic Research Program of China (Grant number 2019YFE0126600); the Major Project of Science and Technology of Henan Province (Grant number 201400210300); the Key Scientific and Technological Project of Henan Province (Grant number 212102210496); the Key Research and Promotion Projects of Henan Province (Grant numbers 212102210393; 202102110121, 202102210368, 19210221009) and Kaifeng science and technology development plan (Grant number 2002001); National Natural Science Foundation of China (62176087) and Shenzhen Science and Technology Innovation Commission (SZSTI)-Shenzhen Virtual University Park (SZVUP) Special Fund Project (2021Szvup032).

**Institutional Review Board Statement:** Not applicable.

**Informed Consent Statement:** Not applicable.

**Data Availability Statement:** The following are available online at http://www.escience.cn/people/gongcheng/NWPU-VHR-10.html.NWPU VHR-10 dataset, accessed on 10 May 2021; www.escience.cn/people/gongcheng/DIOR.html, Dior dataset, accessed on 7 October 2020.

**Conflicts of Interest:** The authors declare no conflict of interest.

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
