# Peer review of "RepDarkNet: A Multi-Branched Detector for Small-Target Detection in Remote Sensing Images"

_ijgi, doi:10.3390/ijgi11030158_

Round 1

Reviewer 1 Report

Target detection is really a typical task in image processing and remote sensing. The authours have proposed the approach based on NN training and show that the proposed NN outperforms some known counterparts. Quite intensibe study and thorough analysis have been done. Due to this, the paper is, in general, of appropriate level. However, to my opinion, the paper canbe improved if: 

1) the authors will clearly explain what doe they mean by remote sensing and the corresponding images; what is a supposed carrier of a sensor, does this sensor acquires a standard RGB image of these are multispectral images where certain sub-bands are used? 

2) the authors will give initial assumptions on sensor spatial resolution and car size expressed in minimal and maximal number of pixels; what about lorries, are they detected well, what types of objects are falsely detected? 

3) the authors will provide initial assumptions on image properties as is noise present or not, is blur of moving targets possible or no; what about the ipossible nfluence of image compression? 

4) the authors will discuss how general is the trained detector; suppose a system has component images for other central frequencies then have we to retrain the network? Is it possible to apply additional training? 

5) What other limitations of the proposed approach do You see? Can it be used for video frames and real-time object detection?  

Reviewer 2 Report

The manuscript proposes the combination of recently published techniques for target  detection (cars) in remote sensing images. Unfortunately, the explanation of some of those techniques is not satisfactory. 

The "Rep" in the title and used throughout the manuscript is never defined. The RepVGG paper clearly states in their abstract that "Such decoupling of the training-time and inference-time architecture is realized by a structural re-parameterization technique so that the model is named RepVGG". Such re-parametrization is not clearly described in this manuscript. Equations 2-8 don't seem to match the simple explanation given in the RepVGG paper that describes that 1x1 convolutions are used when the dimensions are not matching for the residual. 

CSP is used throughout the manuscript, but never defined or explained. 

I could not open the DIOR link. The NWPU-VHR refers to a Chinese website that does not seem to be the main reference. 

More information about the training would be helpful (e.g., optimizer, batch size, how long it takes to train with author's hardware, etc.)

Minor comments

Figure 1 caption could include information about the source. 

Line 48: "In many depth models" - what are "depth models"?

Line 48: "often a shared network" - I don't understand the sharing here. 

Line 66: GIoU was not defined (neither IoU)

Line 75: GAN was not defined

Lines 77-78: is computational supposed to be convolutional?

Line 123: "as in Figure. 1": Figure 2?

Figure 2: There are three MaxPool in the SPP box, but no explanation about their difference. The figure, in general, is not very easy to understand. The purple-ish arrows in the main panel are not explained. The residual block schematic is missing the addition information. 

Figure 4 uses "concate" whereas figure 2 uses "Concat"

In figure 4, the dimensions of C1 are represented as larger than the dimensions of C2. How are they concatenated? The arrows are missing the operation to make the concatenation possible. 

Line 183: IoU was not defined. 

Line 229: is prevalent supposed to be precision?

Line 233: AP and mAP were not defined. 

Table 1 has root symbols instead of a checkmark symbol.

Table 2 suggest to highlight (e.g., bold) the best performing model for each one of the metrics. 

Figure 5 needs to be modified to include x and y axis labels and remove the unecessary information (e.g., approx. time left, Press 's' to save, etc. 

FIgure 7 yellow ellipses should be described in the caption. 

Author Response

Thank you for your letter and for the reviewers’ comments concerning our manuscript entitled “RepDarkNet: A Multi Branched Detector for Small-Target Detection in Remote Sensing Images” (ID: ijgi-1568620). Those comments are all valuable and very helpful for revising and improving our paper, as well as the important guiding significance to our researches. We have studied comments carefully and have made correction which we hope meet with approval. The main corrections in the paper and the responses to the reviewer’s comments are given as follows:

Reviewer 3 Report

The work is well organized, and introduces an improvement in the object recognition steps from remote sensing images.
The experimental results of this work highlight the improvement obtained, both from images and numerically.
From the point of view of supervised learning networks it is usual to have improvements by increasing the training dataset.

However please improve chapter 4 of the work, it is not clear what is meant by ablation experiment, please clarify this better as it is not clear what you mean.

Author Response

(The authors gave the same response as above.)

Round 2

Reviewer 1 Report

I am mostly satisfied by the corrections done and answers.